# Heatwaves, hospitals and health system resilience in England: a qualitative assessment of frontline perspectives from the hot summer of 2019

Katya Brooks ![ORCID],[1] Owen Landeg ![ORCID],[1] Sari Kovats,[2] Mark Sewell,[3] Emer OConnell ![ORCID] [1]

[1]Extreme Events and Health Protection Team, UK Health Security Agency (UKHSA), London, UK
[2]Department of Social and Environmental Health Research, London School of Hygiene & Tropical Medicine, London, UK
[3]National Emergency Preparedness Resilience and Response, NHS Improvement, London, UK

**Correspondence to**
Dr Emer OConnell;
emer.publichealth@gmail.com

## ABSTRACT

**Objective** To critically assess the impacts of very hot weather on (i) frontline staff in hospitals in England and (ii) on healthcare delivery and patient safety.

**Study design** A qualitative study design using key informant semi-structured interviews, preinterview survey and thematic analysis.

**Setting** England.

**Participants** 14 health professionals in the National Health Service (clinicians and non-clinicians, including facilities managers and emergency preparedness, resilience and response professionals).

**Results** Hot weather in 2019 caused significant disruption to health services, facilities and equipment, staff and patient discomfort, and an acute increase in hospital admissions. Levels of awareness varied between clinical and non-clinical staff of the Heatwave Plan for England, Heat-Health Alerts and associated guidance. Response to heatwaves was affected by competing priorities and tensions including infection control, electric fan usage and patient safety.

**Conclusions** Healthcare delivery staff experience difficulty in managing heat risks in hospitals. Priority should be given to workforce development and strategic, long-term planning, prevention and investment to enable staff to prepare and respond, as well as to improve health system resilience to current and future heat-health risks. Further research with a wider, larger cohort is required to develop the evidence base on the impacts, including the costs of those impacts, and to assess the effectiveness and feasibility of interventions. Forming a national picture of health system resilience to heatwaves will support national adaptation planning for health, in addition to informing strategic prevention and effective emergency response.

## INTRODUCTION

The frequency, intensity and duration of heatwaves are increasing in the UK due to anthropogenic climate change.[1–3] The relationship between human health and temperature is well established; heatwaves are already a public health concern and are considered a risk to national security.[4–8] Evidence from the UK's Third Climate Change Risk Assessment (CCRA3) suggests the heatwaves remain an

---

**STRENGTHS AND LIMITATIONS OF THIS STUDY**

⇒ This study explored the experiences of National Health Service employees during hot weather in geographically dispersed healthcare facilities across England in summer 2019.

⇒ Participants interviewed included both clinical (n=4) and non-clinical (n=10) staff and provided a range of perspectives on how heatwaves affect healthcare delivery.

⇒ The clinicians interviewed represented a limited range of clinical specialisms and future studies should look at how heatwaves affect a wide range of clinical care types.

⇒ Interviews were conducted after a heatwave (England experienced three level 3 Heat-Health Alerts in 2019).

⇒ Interviews conducted via telephone or Skype may have limited participants' ability to communicate their experiences and interviewers' ability to interpret them.

---

under-managed risk, and will impact population health and health system delivery. Thus, heat is a priority risk for urgent action for England.[9] Healthcare infrastructure in England is generally not designed to cope with extreme heat and air conditioning is not routinely installed.[10] An estimated 90% of hospital buildings are vulnerable to overheating[11] and National Health Service (NHS) estates are at risk of high indoor temperatures (overheating) even during moderately warm summers[12]; temperatures in some wards can exceed 30ºC even when external temperatures are 22ºC.[13] Existing standards for healthcare premises recommend temperatures from 18ºC to 28ºC in general wards and 18ºC to 25ºC for more sensitive areas, such as birthing and recovery rooms.[12] In 2019–2020, there were 3600 instances of overheating above 26ºC reported in NHS Trust buildings in England.[14] In addition to concerns about patient safety, heat is an occupational health

hazard and can cause discomfort and harm to staff as well as affect productivity.[14–16]

Summer 2019 was warmer than average; with the highest daily temperature recorded in England (38.7°C in Cambridge) until temperatures were broken in the extremely hot summer of 2022.[17] The 2019 heatwave across Europe was made worse by anthropogenic climate change: temperatures were 1.5°C–3°C hotter than they would have been without human influence.[18]

A total of 15 level 2 or level 3 Heat-Health Alerts were issued across all regions of England in June, July and August 2019 based on the thresholds for the Heatwave Plan for England.[19] Public Health England (PHE) estimated that the excess mortality caused by the three heatwaves (level 3 alerts) in 2019 was approximately 892 deaths, the majority of which were in older persons.[20] However, there are no published studies of impacts on hospital admissions. PHE syndromic surveillance showed increases in emergency attendances for heat/sun stroke in line with these periods of hot weather.[21] The Heatwave Plan[19] aims to raise awareness of the health consequences of hot weather and describes what healthcare professionals can do to prepare and plan for hot weather along with actions to take in response to a heatwave. Actions for frontline staff in primary and secondary care include: identify and check on high-risk individuals and raise awareness of heat illnesses and their prevention among clients and carers; identify or create cool rooms/areas where temperatures can be maintained below 26°C and regularly monitor and record indoor temperatures and; ensure business continuity plans are in place and implemented as required.[19]

It has become a priority for the health system to better prepare for future heatwaves, as shown with the inclusion of severe weather response to the NHS Core Standards for Emergency Preparedness, Resilience and Response (EPRR).[22] However, there is limited awareness that hot weather affects delivery of care and patient safety, and limited existing observational evidence about how healthcare settings manage episodes of hot weather. Anecdotal evidence for impacts has been reported during past heatwave events in England.[3 23 24] This paper addresses an important gap in the evidence by exploring the experiences of staff during hot weather in healthcare facilities in England in summer 2019 with a focus on the impacts on health service delivery, staff welfare, and patient safety.

## METHODS
### Study design
A qualitative approach was conducted to collect in-depth data on the experiences of staff during hot weather in healthcare facilities in England in summer 2019 using a preinterview survey and semi-structured interviews.

### Sampling strategy
Participants were identified using snowball, purposive sampling to include those with a broad range of clinical and non-clinical expertise from healthcare facilities across England. Initial approaches were made to participants of the annual PHE heatwave seminar and colleagues identified by NHS England. Sample size was determined by applying the concept of saturation, with suggested individuals being invited and interviewed until no new themes or issues emerged, within the scope of available resources.

Recognising the differences in temperatures experienced across England, a geographical spread of participants from across the country was sought. It is important to emphasise that the summer of 2019 was warm but not the extremes in temperature since seen in England. The study intentionally aimed to explore the experiences of the healthcare workforce dealing with overheating in healthcare estates as a routine summer issue, rather than their response to more unusual extreme heatwave conditions.

### Data collection
Participants were invited by email to a one hour telephone/Skype or face-to-face interview, depending on their geographical location, availability and preference. On acceptance, they were invited to complete a short preinterview online questionnaire in SelectSurvey software, covering actions related to each Heat-Health Alert level (level 2 and level 3) from the Heatwave Plan to inform discussions during the interview. All participants provided written consent prior to participation.

### Data collection instruments and technologies
Semi-structured interviews provided an open framework to engage in a guided discussion with participants on their experiences, challenges, learning and reflections of the period. An interview protocol and topic guide, based around a schedule of common questions, was piloted and employed to ensure consistency and enable comparison across interviews. Interviews comprised guided discussion on the following topics: background information; heatwave planning, preparing and alerting; impacts on participants, their colleagues and working environment; impacts on patients; and recommendations for building future heatwave resilience.

### Units of study
Fourteen semi-structured interviews were completed by key informants in hospital settings comprising four clinical and ten non-clinical employees of the NHS (including estates and facilities and EPRR professionals) (table 1) between October 2019 and January 2020.

### Data processing
For accuracy and transparency, interviews were recorded and transcribed verbatim, and all participants given the opportunity to review their interview transcripts. Each participant was assigned a participant number to ensure anonymity.

### Data analysis
The dataset was thematically analysed and coded using EPPI-Reviewer (Evidence for Policy and Practice Information) software and a coding framework constructed

**Table 1** Breakdown of interviews by interview and role type

| Code | Role type | Region |
|---|---|---|
| Non-clinical 1 | Non-clinical (EPRR) | London |
| Non-clinical 2 | Non-clinical (critical care) | London |
| Clinician 1 | Clinician (acute medicine) | West Yorkshire |
| Non-clinical 3 | Non-clinical (NHS) | National/London |
| Non-clinical 4 | Non-clinical (EPRR) | London |
| Clinician 2 | Clinician (acute medicine) | Liverpool |
| Clinician 3 | Clinician (junior doctor) | London |
| Non-clinical 5 | Non-clinical (EPRR) | London |
| Non-clinical 6 | Non-clinical (EPRR) | Berkshire |
| Non-clinical 7 | Non-clinical (EPRR) | Kent |
| Non-clinical 8 | Non-clinical (estates and facilities) | London |
| Clinician 4 | Clinician | Manchester |
| Non-clinical 9 | Non-clinical (EPRR) | London |
| Non-clinical 10 and Non-clinical 11 (interviewed together) | Non-clinical (EPRR) | Kent |

EPRR, emergency preparedness, resilience and response; NHS, National Health Service.

through an iterative process that was both inductive and deductive, and trialled and agreed by the research team.

### Patient and public involvement

The Health Protection Research Unit in Environmental Change and Health undertakes public involvement and engagement (PPI/E) for its research. Although the PPI/E group 'Planet' was not established at the time this research was undertaken, there have been two workshops on heatwaves and health to plan further research and communicate results effectively to the general public.

### RESULTS

Thematic analysis identified the following key themes: impacts of heat on individual health and well-being (occupational health, and patient health and safety); impacts on healthcare provision; barriers to service delivery; heatwave planning; and consideration of adaptation to climate change.

### Impacts of heat on staff and patients

The majority of participants reported experiencing overheating in healthcare buildings during the summer, with adverse effects on themselves and other staff and patients. Clinical and non-clinical respondents reported significant impacts of high temperatures on both staff and patients in terms of feeling uncomfortable, tired, stressed, unable to cope, less efficient and observing distress in patients.

> It had a huge impact on staff, but also on the patients as well, especially on the wards … some of them were acutely unwell at that time and exacerbated by very uncomfortable conditions. (Non-clinical 14)

> I remember … running around trying to get fans for the patients but the nurses … not having anything at the nurses' station, and people were generally just hot

and stressed and looked uncomfortable. I remember one time we abandoned the handover because we just didn't cope in that room. (Clinician 2)

Some facilities may be particularly prone to overheating due to factors in addition to building design. One EPRR manager for a psychiatric unit reported that:

> Especially for those patients … who are detained under the Mental Health Act and aren't allowed, or it's not safe for them to leave any of the wards …. some of the windows can't be opened for obvious reasons …. I know there was a few issues, understandably, family members were getting extremely distressed with staff saying, you know, this is unacceptable, [and I] totally understand it. … you have some kind of frustrations there amongst patients. We could see a rise in disruptive behaviour on the wards. (Non-clinical 9)

A high level of goodwill for maintaining patient care in the face of challenging working conditions on the wards was recognised by frontline clinicians: 'People did kind of pull together in adversity and everyone tried to keep the whole thing working' (Clinician 3); 'That's probably the motto of the NHS: "just get on with it"' (Clinician 4), as well as non-clinical staff: 'Our staff are really adaptable: whether it's hot or cold, they're really adaptable and the patients are at the heart of whatever they do and they check if our patients are ok in their own homes. Our staff are fantastic, they are brilliant' (Non-clinical 10/11).

### Increases in health service use, particularly for vulnerable adults

Respondents reported an increase in admissions on hot days: '… you definitely notice that it's busier' (Clinician 2). Patients were reported to present with physical

symptoms of heat exhaustion, dehydration, cardiovascular and respiratory conditions, as well as 'injuries related to people doing kind of [outdoor] activities that you do in hot weather; so cycling, going on motorbike riding, drinking alcohol, things like that ….' (Clinician 4).

A specialist in acute medicine, reflected: '[As well as] elderly people getting dehydrated, getting low blood pressure and having to come in because of fainting episodes and things like that and vulnerable groups that aren't able to rehydrate themselves, ask for water or reach the water because they've got dementia, or are very frail ….' (Clinician 4).

In addition to clinical risk factors, clinical respondents reported patients' housing or socioeconomic status as putting them at greater risk in hot weather:

[With one patient I remember] it was like late evening, so you can't speak to the GP and I remember kind of thinking, "My god, we're going to send him back to his [hot] flat which sounds like it's really precipitating continuous seizures and I don't know what kind of longer-term effects that's having …." (Clinician 3)

There were other concerns about discharging patients reported. Only two respondents, both EPRR managers, reported protective procedures in place:

… [for] patients who are being discharged home during a Level 3 heatwave … we have a discharge proforma that gets filled in …. It highlights…these people will be at risk if they're on their own in a top floor flat with drugs that inhibit sweating, or inability to be walking around and getting drinks and keeping cool, for example. And then we would give those people a copy of the PHE [Beat the Heat] leaflet. Or is it safe to actually get them at home with some kind of social services input? (Non-clinical 7)

Social isolation was also raised as a compounding factor: '… people who were more socially isolated, perhaps living on their own, just managing most of the time, then weren't managing' (Clinician 1).

### Impacts of heat on facilities and equipment

Significant impacts on medical equipment and facilities that led to impacts on healthcare provision were reported. In relation to medical equipment, an EPRR manager experienced failure of medication storage facilities in a psychiatric unit: 'Some of our medication, especially the anti-psychotic medication, needs to be kept at a specific temperature …. I believe on a couple of occasions the actual fridge which was containing some medication had bizarrely overheated, or just broke down' (Non-clinical 9). Two respondents reported issues due to server rooms overheating and as a result: 'they lost quite a lot of IT aspects, including the ability for all their different systems to talk to each other' (Non-clinical 3).

An estates and facilities manager reported high indoor temperatures in critical care units, operating theatres and rooms with MRI scanners which caused disruption to patient care, from loss of beds (which became unusable) and loss of equipment (scanners became unusable).

The intensive care unit was probably our worst area, and we struggled with some of our [MRI] scanners to keep them cool in certain periods as well, and that would have impacted on patient care. The paediatric scanning was suspended for periods; it was too unpleasant for the young ones to go through it. (Non-clinical 8)

This view was echoed by an EPRR manager from the same hospital:

Probably because the heat was rising and it just put enormous pressure on [our air conditioning unit] but the contingency didn't work, which meant that … half of the ITU beds were lost or they were overheating …. We had to look at evacuating the critical care area because it was just getting too hot …. We moved some vulnerable out of those areas …. There were theatre suites that … because the air conditioning was working so hard, they got full condensation so there was literally water running down on the walls and they had to stop operating … just to dry the walls, then deep clean it and then continue with the theatre list. (Non-clinical 5)

Critical care facilities have a large number of air conditioners. These two respondents also reported that the increased demand during the hot weather caused a power failure which exacerbated indoor temperatures due to lack of air conditioning and also caused problems for refrigeration services as well as the mortuary: '…. the mortuary fridges failed. In response, the Estates and Facilities Team 'converted a container for a temporary mortuary' (Non-clinical 8). As a direct consequence of these concurrent events and their cumulative impacts on service delivery and patient safety, this hospital declared an internal incident due to the heatwave.

### Barriers to service delivery in a heatwave

Several barriers to delivery of care in a heatwave were identified by respondents. These have been categorised in terms of preparedness and response to Heat-Health Alerts, built environment and managing indoor temperatures, and behaviours.

#### Preparedness and response to Heat-Health Alerts

Lack of time and capacity to respond to high temperatures due to competing priorities for frontline staff were reported across clinical and non-clinical respondents:

… Often when these things happen you're just so, so busy and it sometimes feels like you go from one crisis to another, that you don't actually have an awful lot of time to step back and say: "Right. What can we do differently?" (Clinician 2)

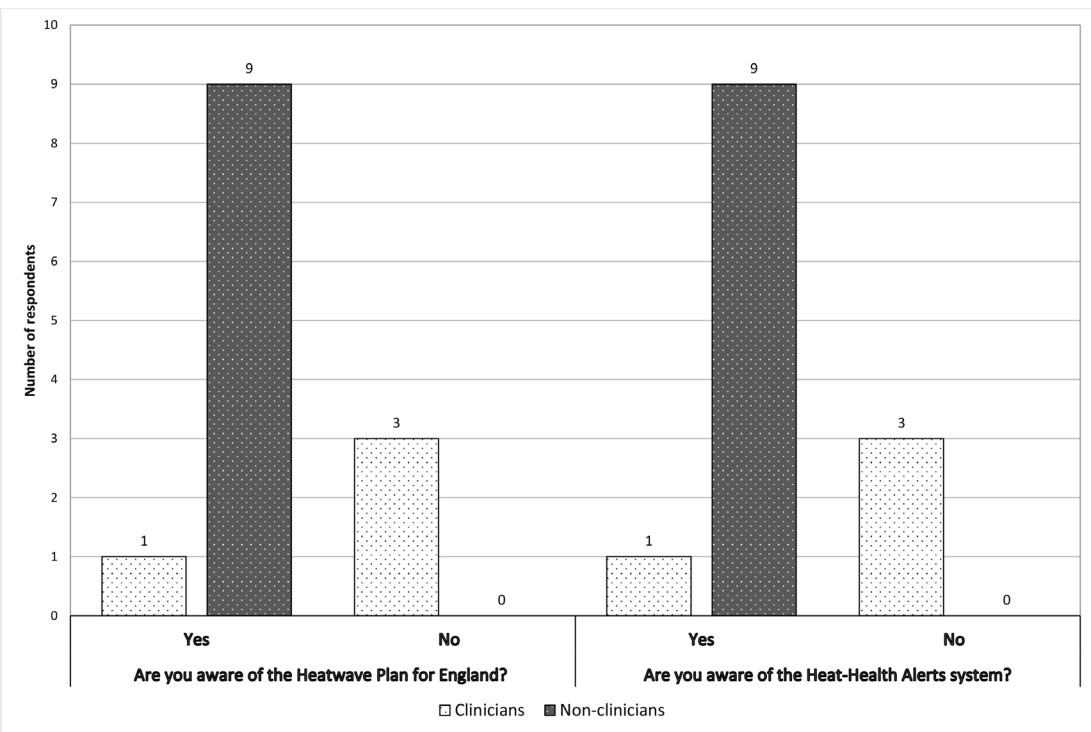

**Figure 1** Pre-interview questionnaire responses: awareness of the Heatwave Plan for England and the Heat-Health Alerts system.

Data from pre-interview questionnaires indicate a greater lack of awareness of the Heatwave Plan for England and Trust-level plans and Heat-Health Alerts among clinical respondents as compared with non-clinical respondents (see figure 1).

The majority of respondents reported that they received or cascaded Heat-Health Alerts by email. However, no clinician could recall receiving the alerts: 'Realistically as a [Junior Doctor] you're probably not logging on to your emails, your NHS emails, that often' (Clinician 3). A reality recognised by some EPRR practitioners: '[Email is] the best way to communicate really on heatwave alerts …. But, will they look at it? No way, not really' (Non-clinical 5) and 'It's a kind of, they'll see on the news or something' (Non-clinical 9).

Across clinical respondents, a significant difference was observed in levels of awareness of the heat-health risks in general, in comparison with winter and cold weather:

Hot weather is never something that I've heard mentioned. Hot weather is almost one of those things you just get on and deal with … Whereas all the time, even in the middle of summer, we're going to a wrap-up last winter meeting and then you go to a, what are we going to do next winter meeting. (Clinician 2)

The majority of EPRR professionals interviewed identified the need for strategic, long-term planning, preparation and investment across the system to ensure healthcare resilience to all extreme weather hazards: 'The key thing for me, is about relationship building with [Estates and Facilities], so we will work closely with them

all year round, not just when we suddenly get a heatwave alert' (Non-clinical 7).

We need to be much more prepared …. it's kind of frustrating really that we kind of deal with it when we're in the midst of it, you know …. So I think the lessons that we've learnt is almost we need to prepare from the turn of the year. (Non-clinical 9)

Sometimes it's a challenge to get people to start thinking long term, and investment, money will always come down to being part of the issue …. (Non-clinical 1)

[The Heatwave Plan] gets circulated [annually] before our heat season starts and for people to say, please remember this. But the reality is that the majority of people have never opened it up and they'll kind of get almost kind of hit by it on the day when it gets hot. It's unfortunate it's a very reactive plan, rather than concentrating on this year-round. (Non-clinical 5)

### Challenges of managing indoor temperatures

Challenges of managing indoor temperatures often arise from the design of NHS buildings, which vary in the insulation and ventilation characteristics. Indoor temperatures are also increased by indoor heat sources (computers and equipment). Adaptation measures are limited by a lack of financial investment in the management of healthcare estates. One EPRR manager reflected: 'The NHS is ageing and there's very little financially that we can do about it ….' (Non-clinical 5). A financial

barrier reported by three non-clinical respondents stems from outsourced ownership of healthcare estate buildings, property management and maintenance leading to lack of control in making decisions to manage internal temperatures:

> The majority of buildings are not owned by us … and we are tenants, and that's where we tend to get problems: There's normally quite a lot that's out of our control as it's a property services building so we can't necessarily do what we'd ideally want to do …. It would be nice to take control of own destinies. (Non-clinical 10/11)

There is some evidence that new buildings are more adapted to hot weather. An EPRR manager responsible for two sites commented that:

> … Our [new] site, which is designed to kind of withstand heatwaves, we didn't have to activate any contingencies there at all … whereas at [the older site] … we definitely see that increase in [internal] temperature here …. we were getting … multiple incident forms every day in the summer. (Non-clinical 7)

Infection prevention and control (IPC) policy was identified as a significant barrier to the use of electric fans and to a lesser extent, air conditioning units in ward settings by all clinical and the majority of non-clinical respondents: 'Infection control is one of the overriding mantras everywhere' (Clinician 1). Six respondents reported complete bans on electric fan usage in clinical areas 'We were just told that due to infection control we weren't allowed to use [electric fans] for patients' (Clinician 4); 'They wouldn't allow fans because they said it was a health and safety risk' (Clinician 3); '… a clinical alert came out regarding … the fans … and said that … they're not to be used in clinical areas …. Because of infection control … and that came out just before the heatwave, you know, which made it quite difficult' (Non-clinical 9). Others reported fans being used as expected, with inpatient areas prioritised: 'I remember people … running around trying to get fans for the patients and the nurses sort of not having anything at the nurses' station' (Clinician 2). The difficulty of ensuring enough fans were available to provide sufficient cooling during heatwaves was cited: 'We had fans …. So all that was kind of in place, there just wasn't enough of it …. It's impossible to test like how many fans does it take to cool down this ward when it gets to 34 degrees' (Non-clinical 5).

Two EPRR managers stated that, due to financial and IPC constraints, the decision was made in their hospitals to annually dispose of electric fans and buy replacements to benefit patient care, if not the environment: 'Is it better to cool down the infectious patients with the fan, you know, and then chuck it, the fan, afterwards if you can't clean it, or let the poor man or woman just melt?' (Non-clinical 5) and, 'This is not great for the environment but it's cheaper actually just to bin them and buy new fans' (Non-clinical 1).

Most EPRR respondents reported annual procurement of portable air conditioning units: 'Where it's possible we bring in mobile air con units, but that's obviously not an ideal response' (Non-clinical 4).

> We can't purchase something portable and keep them in a cupboard, it's not a cost-effective way to do this. We will have to use an external company who can store them; you know, maintain them and test them and will bring them on site and hire them for that period of time. (Non-clinical 5)

Two respondents reported instances where air conditioning could not be used across their sites: 'A lot of our corporate settings had the portable air conditioning units that could be used, but of course they can't be used in clinical settings because of infection control' (Non-clinical 9).

> There were five … inpatient areas in old buildings that didn't have air conditioning installed, couldn't— either for the size or the location of the ward, or the complexity of the patients' [needs] …. We had outpatient areas where there could be children; you can't put any kind of air conditioning unit because of the risk of kiddies touching them. The only place with air conditioning in some areas would be in the middle of the nurses' station because of the risk of trip hazards, of the wires and of course, to elderly patients. (Non-clinical 5)

Participants called for increased clarity of infection control guidance at a national level as 'different trusts' infection control people say different things' (Non-clinical 7) in order 'to demystify everything. For example, that air conditioning can or can't be used due to legionella' (Non-clinical 6).

Participants questioned the feasibility of actions outlined in the Heatwave Plan, to create cool rooms and for checking indoor temperatures.

> … Every bed is almost always full and there isn't necessarily extra space in the hospital to have a cooler area or the infrastructure to cool an area down …. it would be very challenging to implement actually, and I think … it would be difficult to kind of move someone out of the cool area into the hotter area in exchange for another patient, because you're subjecting them to a bit more discomfort and possibly risk as well. (Clinician 4)

> … What if your coolest room is not near your nursing station, but you're going to put your vulnerable patients in there? … Is it right that we put a vulnerable patient further away just because it's a bit cooler than anywhere else? (Non-clinical 1)

### Role of adaptive behaviours

Heat risks can be reduced by changes in behaviour, in both frontline staff and patients. All participants reported lack of knowledge as a major challenge to adopting cooling

strategies in health facilities, such as closing curtains or blinds, moving patients, etc.: '… There doesn't appear to be any plan on it, but I think it was down to … nurses thinking we need to keep this patient in the shade' (Clinician 1).

Workplace culture and attitudes were raised in relation as a barrier to staff drinking water and keeping hydrated while working on the ward due to fear of appearing unprofessional. One EPRR manager sought to address this problem:

> We made it very clear to staff that we wanted to see them carrying water around, we wanted to see them drinking on duty, just to dispel all this old nonsense about, "oh you shouldn't be seen drinking on duty, it's unprofessional". Actually, we said: "It is professional, it's looking after yourself". (Non-clinical 7)

## DISCUSSION

Our findings provide insights regarding the range of challenges experienced by staff and patients during hot weather in settings with limited processes and interventions in place to manage these conditions. Previous studies of individual hospitals and wards have identified factors related to the built environment and management of indoor temperatures (eg,[3 23]), our findings broaden this perspective and identify wider considerations such as increased service demand, discomfort of patients and staff, and rooms and equipment becoming unusable or failing, leading to disruptions in service. It also highlights the importance of staff behaviours, clinical practice and organisational processes during heat episodes and how these factors may affect staff and patient well-being as well as healthcare delivery.

Barriers to care delivery associated with preparedness and response were also identified: capacity restraints, difficulties in identification and prioritisation of risk mitigating actions, gaps in awareness of heat-health risks and understanding of actions to take on receipt of an alert were all cited.[25] These findings align with those of the *Evaluation of the Heatwave Plan for England*'s 2019 national survey of nurses in hospital, community and care home settings which reported that 'many frontline staff, including nurses surveyed, reported to be unaware of any local heatwave plans, and unfamiliar with the Heatwave Plan for England guidance … many reported taking few or none of the recommended Heatwave Plan actions during an alert'."[26]

This research provides important insights on operational barriers to action, particularly in relation to use of electric fans as a cooling strategy and IPC concerns. Implementation of individual cooling interventions should form part of a whole systems approach that considers a hierarchy of cooling interventions, with active cooling as a last resort. The approach to cooling for any indoor environment should take account of the effectiveness, cost, feasibility, ease of use and sustainability of the different options in that context—in some settings, a combination of options may work best. This study has highlighted that a lack of clarity or consistency of policy on the use of fans and/or cooling systems has resulted in heat mitigations not being used and/or costly work-arounds being put in place due to concerns around IPC. Heat is one of several concurrent risks in these setting and trade-offs may be inevitable, but decisions should be supported by good evidence and this is currently lacking. Further, the routine disposal and replacement of electric fans does not align with net zero or sustainability commitments.

Future iterations of the Heatwave Plan for England should consider the barriers identified in this study and consider the feasibility of preventative action within clinical settings. Improvements in awareness and engagement with the existing Heatwave Plan and associated Heat-Health Alerts are needed, particularly for frontline staff. Efforts on workforce development should focus on empowering and supporting staff to understand and act on Heat-Health Alerts and to embed adaptive behaviours in their daily practice. Additionally, overheating should be considered/addressed within the forthcoming update to the National Adaptation Programme and included in the broader adaptation elements of NHS Green Plans.

For the medium to longer term, the diversity of the challenges identified highlights the need for a comprehensive assessment of the resilience of healthcare estates in England to inform future policy priorities. Ensuring a climate resilient health system will require strategic, long-term planning and investment to address the drivers for overheating risk and the barriers to action both before and during a heatwave event. At a minimum, any investment to improve energy efficiency should also consider adaptation.

### Limitations

This study has several limitations. Interviews were conducted several months after the heatwave event, which may have led to recall bias. It is acknowledged that interviewees were more likely to be those who were more aware and had experienced difficulties during summer 2019. However, this is appropriate due to the aim of the paper to explore the pathways through which heat affects health service delivery, not to provide a complete overview of the total impact. Relatively low engagement from clinicians in comparison with non-clinical NHS representatives may introduce a bias in terms of perspective. Interviews conducted via telephone or Skype may have limited participants' ability to communicate, and interviewers' ability to interpret, their experiences. However, this enabled engagement with participants working in more geographically dispersed locations across England than would otherwise have been possible.

### CONCLUSION

This study is the first to consider the broad range of challenges faced by staff and patients when experiencing hot

weather in healthcare estates in England. Many national public health systems are developing methods, plans and tools to manage climate risks, but adaptation is still largely viewed as an emergency response activity. Further action is needed to improve the climate resilience of the health system, in particular to address overheating in care settings, while committing to reaching carbon net zero by 2040.[27] The qualitative evidence presented here of the experiences of healthcare staff in England demonstrates that there is an issue, and it is not well quantified. Policies that support a climate resilient healthcare system need to be based on good evidence on the extent and scale of the problem. Specifically, additional research is recommended to further develop the evidence base on the impacts, viability of interventions—particularly in relation to a healthcare setting where IPC must also be considered—and current costs associated with heat-related disruption to health service delivery. This should inform a broader assessment of the climate vulnerability of the health system, and future planning assumptions for strategic prevention and emergency response.

**Acknowledgements** The authors wish to thank the stakeholders that participated in this research.

**Contributors** KB and OL contributed equally to this paper. KB and OL designed the study, collected and analysed the data, drafted and revised the paper. MS contributed to the design and implementation of the study. SK contributed to the design of the study. EOC is guarantor for this study, contributed to the design, critically reviewed and finalised the draft paper.

**Funding** This work was supported by the National Institute for Health Research (NIHR) Health Protection Research Unit in Environmental Change (NIHR200909), a partnership between the UK Health Security Agency (UKHSA) and the London School of Hygiene & Tropical Medicine. The views expressed are those of the author(s) and not necessarily those of the NIHR, UKHSA or the Department of Health and Social Care.

**Competing interests** None declared.

**Patient and public involvement** Patients and/or the public were not involved in the design, or conduct, or reporting, or dissemination plans of this research.

**Patient consent for publication** Not applicable.

**Ethics approval** This study involves human participants and was approved by Public Health England Research Ethics and Governance Group: R&D reference 366. Participants gave informed consent to participate in the study before taking part.

**Provenance and peer review** Not commissioned; externally peer reviewed.

**Data availability statement** No data are available. The participants of this study did not give written consent for their data to be shared publicly, so due to the small number of participants and the potential for them to be identified on the basis of their responses, the supporting data are not available.

**ORCID iDs**
Katya Brooks http://orcid.org/0000-0002-8283-7378
Owen Landeg http://orcid.org/0000-0002-2598-4769
Emer OConnell http://orcid.org/0000-0002-9887-5731

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
