## [Reviewer comments · BMJ Open]

ARTICLE DETAILS

TITLE (PROVISIONAL)	Heatwaves, hospitals, and health system resilience in England: a qualitative assessment of frontline perspectives from the hot summer of 2019
AUTHORS	Brooks, Katya; Landeg, Owen; Kovats, Sari; Sewell, Mark; OConnell, Emer

VERSION 1 – REVIEW

REVIEWER	Moon, Jinyoung Seoul National University, Department of Environmental Health Science
REVIEW RETURNED	04-Oct-2022

GENERAL COMMENTS	General comments The reviewer thinks this manuscript is not a scientific article in medical science. This article is a type of exploratory article containing 14 interviews. This manuscript would fit well with policy journals or journals on social science. There is rarely helpful information at this time (September and October 2022) in this manuscript. This summer in 2022 was the hottest summer in England, and many journals in the social science sector are dealing with this subject. There is no hypothesis in this article. Furthermore, scientific validation is limited in this manuscript because this type of article is not based on experimental or statistical principles. The reviewer recommends re-arranging and re-organizing the interview materials. In addition, the reviewer recommends that the authors draw several policy recommendations based on the existing 14 interviews.
--

REVIEWER	Howarth, Candice The London School of Economics and Political Science, Grantham Research Institute on Climate C
REVIEW RETURNED	26-Oct-2022

GENERAL COMMENTS	Many thanks for the opportunity to review this timely paper, which I enjoyed reading. I found it very interesting and well written and have only a few minor suggestions to improve it. - The Introduction could be elaborated slightly to include a rapid overview of the UK policy landscape regarding responses to high temperatures and reference to other studies on this topic (not necessarily an NHS setting) but for example work that has been carried out looking at schools and care home settings and how lessons from those studies could be applied to this paper
--

	 - Could the Introduction also include any statistics on heat-related hospital admissions that summer? - Details on the sampling method for the interviews could be elaborated, specifically outlining how heat may impact clinical and non-clinical staff and hence why they were the focus of this research, whether the geographical locations covered by the interviews were more/less exposed to high temperatures during the heatwave, how much they already knew/understood about heat risk in their settings, their level of seniority (and whether this would affect their receiving information related to the heat warnings) -Details on how interviewees were recruited/approached would be useful to include and discuss whether this could have led to selection bias -Details of what the interview discussions covered is required - It would be interesting to add a rapid analysis comparing responses/views of the interviewees according to whether they were clinical/non-clinical - The discussion would benefit from discussion on the implications of the limitations of the study and the extent to which the interview findings can be seen as representative. The small sample size is problematic, particularly the balance of clinical vs non-clinical interviewees so this also needs wider discussion of overcoming any issues relating to this - The first reference should be 'Intergovernmental' Panel on Climate Change, not 'International'
--	--

REVIEWER	Bikomeye, Jean Medical College of Wisconsin, Institute for Health & Equity
REVIEW RETURNED	29-Oct-2022

GENERAL COMMENTS	Review Assignment for bmjopen-2022-068298 Title: Heatwaves, hospitals, and health system resilience in England: a qualitative assessment of frontline perspectives from the hot summer of 2019 Summary of the Paper: The paper attempts to critically assess the effects/impact of heatwaves (very hot weather in the summer of 2019) on frontline staff in hospitals in England, and on health care delivery and patient safety. Authors conduct qualitative analysis with key informants interviews and use thematic analysis to draw insights, views, or perceptions. Authors identified 14 employees of NHS using appropriate sampling strategy (clinicians, non-clinicians including facilities 16 managers and emergency preparedness, resilience, and response (EPRR) professionals). Along with these 14 participants, authors obtained, from thematic assessment, these following ideas: impacts of heat on individual health and wellbeing (occupational health, and patient health and safety); impacts on healthcare, provision; barriers to service delivery heatwave planning and consideration of adaptation to climate change. After thematic analysis, authors found that staff and patients experience overheating in healthcare buildings during summertime and as result. They note important differences between older
---

	building (having more issues) and newer buildings (having less issues due to heatwaves). Authors also found that during hot days, there were increase in admissions, particularly for vulnerable adults. Authors found that there were issues (or concerns) from providers in discharging patients who would go back to their homes (that might also be very hot and putting patients at increased risk for heatwaves again even if they follow necessary protocols including issuing a 'leaflet' with heatwaves information and actions to take), and social isolation. Authors found that there were effects on hospital facilities and equipment. Authors also identify barriers to health care delivery of services during heatwaves. While authors have overall done an excellent job, I have a few comments that should be addressed before the paper gets published. I will add overall comments and sections specific comments. Comments for paper's Introduction: Regarding the in-text citations, I suggest the authors should revise the positioning of these citation. Observing the line 44, citations (1, 2, 3) come after the point. The point should come after the citations (unless BMJ Open has an exception). With this, I suggest that the authors should revise all in-text citations: Check lines 45, 48, 52, 60 and so on, and ensure correct in-text citations. Though authors revealed the aim of the study, I feel like they do not show (or at least clearly name) the contribution of their to the literature on adverse effects of heatwaves (and other extreme weather events) in disrupting the health systems. I think we already know that heatwaves negatively impact clinical staff and patients of course (read this paper for example: https://www.ncbi.nlm.nih.gov/pmc/articles/PMC9013542/). But the paper had some very good stories that highlight the vulnerabilities of existing health systems during crisis. It has a big room in the literature, but authors MUST name what the contribution is. Is the focus on UK (England and perception of clinical staff the innovation)? I also wonder if introducing a "Climate and Health" framework might add more flavor to the paper. I suggest authors to look at these two papers: (1) https://www.ncbi.nlm.nih.gov/pmc/articles/PMC7967605/, and (2) https://catalyst.nejm.org/doi/full/10.1056/CAT.21.0454, and see if the views expressed in the papers might be thoughts for systems resilience as they name it later in their discussion. I think the incorporation of readings form the above suggested papers, might add more flavor in authors' new sentence about contribution to the literature by comparing to most recent publication in the field. I also spotted two abbreviations that I did not know what they meant: EPPI, and NHS. There might be others that I might. I think it should be a better practice to define abbreviations when their first appear in the paper. Otherwise, I think is very good and states the issue and purpose of the paper. Methods of the paper Authors clearly explained the methods; and I can follow through; but I hope one of reviewers is an excellent qualitative methodologist and be able to judge the soundness of the methods. I do not have any objection, but I highly suggest inviting a qualitative methodologist to ensure that the soundness is solid. Results/Discussion/Conclusion:
--	--

	Authors expound well the findings/results in a thematic way. Again, a qualitative methodologist would be better positioned to advice on best reporting. Regarding the discussion, though in some few paragraphs, some evidence were indicated to back up the results, the discussion section is not research/evidence-heavy. It does not have a lot of evidence (citations) to support claims that authors are making. I suggest that authors should revise the discussion part and support heavily their claims with recent sources.
--	---

VERSION 1 – AUTHOR RESPONSE

Reviewer: 1

Dr. Jinyoung Moon, Seoul National University

Comments to the Author:

Review comments

Title: Heatwaves, hospitals, and health system resilience in England: a qualitative assessment of frontline perspectives from the hot summer of 2019

General comments

The reviewer thinks this manuscript is not a scientific article in medical science. This article is a type of exploratory article containing 14 interviews. This manuscript would fit well with policy journals or journals on social science. There is rarely helpful information at this time (September and October 2022) in this manuscript. This summer in 2022 was the hottest summer in England, and many journals in the social science sector are dealing with this subject. There is no hypothesis in this article. Furthermore, scientific validation is limited in this manuscript because this type of article is not based on experimental or statistical principles. The reviewer recommends re-arranging and re-organizing the interview materials. In addition, the reviewer recommends that the authors draw several policy recommendations based on the existing 14 interviews.

AUTHORS RESPONSE

There is limited evidence on the experience of the workforce in the UK and how they experience hot weather and the challenges they face delivering services. Circumstances are specific given no air conditioning, aging infrastructure, and only recently experiencing temperatures at this level. Qualitative research is generally used to establish themes and to enable further quantitative evidence to be gathered. This study is part of a wider research within the HPRU to establish the situation, both in breadth and depth. The study provides important insights into the range of experiences of the workforce and their ability to safely manage patient care, including patient safety. This evidence has supported the development of additional research to better understand these issues in more detail.

2019 was a warm summer but not exceptional. The heatwave of summer 2022 was an extreme event that will require additional investigation. The purpose of this study was to better understand the experience of 'routine' hot weather that nonetheless creates specific issues for healthcare services.

Additional text has been added to the Conclusions to include policy recommendations.

+++++

Reviewer: 2

Candice Howarth, The London School of Economics and Political Science

Comments to the Author:

Many thanks for the opportunity to review this timely paper, which I enjoyed reading. I found it very interesting and well written and have only a few minor suggestions to improve it.

The Introduction could be elaborated slightly to include a rapid overview of the UK policy landscape regarding responses to high temperatures and reference to other studies on this topic (not necessarily an NHS setting) but for example work that has been carried out looking at schools and care home settings and how lessons from those studies could be applied to this paper

- There is a wide range of policy that relates to managing heat risk, however, this research project is focussed on the impact of heatwaves on health workforce and health service delivery, which is most relevant to the readership of the BMJ. We do not feel it is relevant to include detailed information about operational responses in other settings in this paper. We do refer to the Heatwave Plan for England which is the main public health strategy for addressing heatwave in England.

Could the Introduction also include any statistics on heat-related hospital admissions that summer?

- Text amended to reflect that there are currently no published studies of excess hospital admissions or increased demands for other health services during the 2019 heatwave in England. However, the impacts of the heatwave on acute mortality have been published. We also have included information from other heatwave events on acute increases in morbidity and demand for services that is relevant.

Details on the sampling method for the interviews could be elaborated, specifically outlining how heat may impact clinical and non-clinical staff and hence why they were the focus of this research, whether the geographical locations covered by the interviews were more/less exposed to high temperatures during the heatwave, how much they already knew/understood about heat risk in their settings, their level of seniority (and whether this would affect their receiving information related to the heat warnings

- Additional text covering these aspects has been added to the section on sampling

Details on how interviewees were recruited/approached would be useful to include and discuss whether this could have led to selection bias

- Additional text included within the sampling strategy and limitations sections to address this.

Details of what the interview discussions covered is required

- Text added under the subheading 'Data collection instruments and technologies'

It would be interesting to add a rapid analysis comparing responses/views of the interviewees according to whether they were clinical/non-clinical

- The sample size is small (n=14) and within group differences are quite significant so it is not clear what patterns would be discernible or what value this would add. A key finding highlighted in the results is that whilst emergency planners are aware of the Heatwave Plan, clinical staff are generally not aware.

The discussion would benefit from discussion on the implications of the limitations of the study and the extent to which the interview findings can be seen as representative. The small sample size is problematic, particularly the balance of clinical vs non-clinical interviewees so this also needs wider discussion of overcoming any issues relating to this

- Interviews do not need to be representative of the entire workforce, that is a misunderstanding of qualitative methods. Additional text has been added to the Limitations and Discussion sections to clarify the purpose of qualitative exploratory research.

The first reference should be 'Intergovernmental' Panel on Climate Change, not 'International'

- Text and reference have been changed to 'Intergovernmental'

+++++

Reviewer: 3

Mr. Jean Bikomeye, Medical College of Wisconsin

Comments to the Author:

I attached a few minor comments for authors.

Regarding the in-text citations, I suggest the authors should revise the positioning of these citations. Observing the line 44, citations (1, 2, 3) come after the point. The point should come after the citations (unless BMJ Open has an exception). With this, I suggest that the authors should revise all in-text citations: Check lines 45, 48, 52, 60 and so on, and ensure correct in-text citations.

- All citations are in line with BMJ Open guidelines, i.e. a full stop comes before the citation.

Though authors revealed the aim of the study, I feel like they do not show (or at least clearly name) the contribution of their [paper?] to the literature on adverse effects of heatwaves (and other extreme weather events) in disrupting the health systems. I think we already know that heatwaves negatively impact clinical staff and patients of course (read this paper for example: <https://www.ncbi.nlm.nih.gov/pmc/articles/PMC9013542/>). But the paper had some very good stories that highlight the vulnerabilities of existing health systems during crisis. It has a big room in the literature, but authors MUST name what the contribution is. Is the focus on UK (England and perception of clinical staff the innovation)?

- Introduction has been revised to more clearly highlight the specific contribution of this paper, particularly in increasing understanding of the range of challenges experienced in England where temperatures are increasing and healthcare estates are not adapted for heat, nor is air conditioning widely available.

I also wonder if introducing a "Climate and Health" framework might add more flavor to the paper. I suggest authors to look at these two papers:

(1)<https://www.ncbi.nlm.nih.gov/pmc/articles/PMC7967605/>, and (2) <https://catalyst.nejm.org/doi/full/10.1056/CAT.21.0454>, and see if the views expressed in the papers might be thoughts for systems resilience as they name it later in their discussion. I think the incorporation of readings from the above suggested papers, might add more flavor in authors' new sentence about contribution to the literature by comparing to most recent publication in the field.

- Thank you for the two suggested papers. References have been added. There is a wide range of policy that relates to managing heat risk, however, this research project is focussed on the impact of heatwaves on health workforce and health service delivery.

I also spotted two abbreviations that I did not know what they meant: EPPI, and NHS. There might be others that I might. I think it should be a better practice to define abbreviations when their first appear in the paper.

- Thank you. Full text added as follows: Evidence for Policy and Practice Information (EPPI, National Health Service (NHS))

Otherwise, I think is very good and states the issue and purpose of the paper. Authors clearly explained the methods; and I can follow through; but I hope one of reviewers is an excellent qualitative methodologist and be able to judge the soundness of the methods. I do not have any objection, but I highly suggest inviting a qualitative methodologist to ensure that the soundness is solid. Authors expound well the findings/results in a thematic way. Again, a qualitative methodologist would be better positioned to advice on best reporting.

- Thank you for this feedback. Two of the researchers SK and KB are very experienced qualitative researchers and are comfortable with the soundness of methods used.

Regarding the discussion, though in some few paragraphs, some evidence were indicated to back up the results, the discussion section is not research/evidence-heavy. It does not have a lot of evidence (citations) to support claims that authors are making. I suggest that authors should revise the discussion part and support heavily their claims with recent

sources.

- We have redrafted and reformatted the discussion adding additional key references to reflect the wider evidence and context on this topic.

VERSION 2 – REVIEW

REVIEWER	Bikomeye, Jean Medical College of Wisconsin, Institute for Health & Equity
REVIEW RETURNED	07-Jan-2023
GENERAL COMMENTS	My comments have adequately been addressed.